# Excellent Catalytic Performance of ISOBAM Stabilized Co/Fe Colloidal Catalysts toward KBH_4_ Hydrolysis

**DOI:** 10.3390/nano12172998

**Published:** 2022-08-30

**Authors:** Keke Guan, Qing Zhu, Zhong Huang, Zhenxia Huang, Haijun Zhang, Junkai Wang, Quanli Jia, Shaowei Zhang

**Affiliations:** 1The State Key Laboratory of Refractories and Metallurgy, Wuhan University of Science and Technology, Wuhan 430081, China; 2School of Materials Science and Engineering, Henan Polytechnic University, Jiaozuo 454003, China; 3Henan Key Laboratory of High Temperature Functional Ceramics, Zhengzhou University, Zhengzhou 450052, China; 4College of Engineering, Mathematics and Physical Sciences, University of Exeter, Exeter EX4 4QF, UK

**Keywords:** ISOBAM-104, Co/Fe colloidal catalysts, hydrogen generation, KBH_4_ hydrolysis, catalytic activity

## Abstract

Recently, developing a cost-effective and high-performance catalyst is regarded as an urgent priority for hydrogen generation technology. In this work, ISOBAM-104 stabilized Co/Fe colloidal catalysts were prepared via a co-reduction method and used for the hydrogen generation from KBH_4_ hydrolysis. The obtained ISOBAM-104 stabilized Co_10_Fe_90_ colloidal catalysts exhibit an outstanding catalytic activity of 37,900 mL-H_2_ min^−1^ g-Co^−1^, which is far higher than that of Fe or Co monometallic nanoparticles (MNPs). The apparent activation energy (*E*_a_) of the as-prepared Co_10_Fe_90_ colloidal catalysts is only 14.6 ± 0.7 kJ mol^−1^, which is much lower than that of previous reported noble metal-based catalysts. The X-ray photoelectron spectroscopy results and density functional theory calculations demonstrate that the electron transfer between Fe and Co atoms is beneficial for the catalytic hydrolysis of KBH_4_.

## 1. Introduction

Recently, hydrogen has been widely considered as a promising clean energy source to replace the traditional fossil fuels. Chemical hydrogen storage materials have aroused tremendous interest because of their inherent advantages such as high content of hydrogen, no toxicity, low hydrogen releasing temperature, and an easily controllable hydrogen generation process [1,2,3,4,5,6,7,8]. Among those materials, potassium borohydride (KBH_4_) stands out owing to its safe production process, harmless hydrolysis product, low activation energy and enthalpy [9,10,11,12,13,14]. Unfortunately, the low hydrogen production rate of KBH_4_ self-hydrolysis hinders its large-scale practical application.

Many researchers found that metal nanoparticles (NPs) could catalyze the hydrolysis of KBH_4_ and accelerate the generation rate of hydrogen [7,15,16]. For example, Kilinc et al. [7] successfully prepared the Pd complex catalysts for promoting the KBH_4_ hydrolysis. The catalytic activity of the as-prepared catalysts was up to 37,900 mL-H_2_ min^−1^ g-catalyst^−1^. Recently, a series of colloidal metal catalysts were synthesized and used for catalyzing the hydrolysis of KBH_4_ [17,18,19,20]. For instance, Wang et al. [19] successfully synthesized colloidal Co single-atom catalysts for the effective production of hydrogen from KBH_4_ hydrolysis by using ISOBAM (isobutylene-alt maleic anhydride) as a protectant. The synthesized colloidal metal catalysts possess a clearly intrinsic catalytic activity of metal without the influence of support. Besides, those colloidal metal catalysts are stabilized by protective agents and present excellent catalytic activity and recyclability.

It has been widely accepted that the bimetallic catalysts exhibited a high catalytic activity for hydrogen production owing to the synergistic effects between different constituents [21,22,23,24,25]. In detail, the addition of another metal component could modify the electronic structure and then improve the catalytic activity [25,26]. For example, a previous report displayed that the Rh_10_Ni_90_ bimetallic nanoparticles (BNPs) possessed a higher catalytic activity for the KBH_4_ hydrolysis than that of Rh or Ni MNPs [27]. The catalytic activity of the reported Au/Ni BNPs was several times higher than their corresponding monometallic counterparts [28]. In addition, some non-noble metal catalysts (including Fe [29,30,31], Ni [18,32,33], Co [19,34], and Cu [35,36]) attract increasing attention owing to their considerable natural abundance, low cost, and competitive catalytic activity. However, the preparation of bimetallic catalysts with noble-free metal constituents is scarcely retrieved.

Herein, we reported a co-reduction method to prepare the ISOBAM-104 stabilized Co/Fe colloidal catalysts, which were then used for the hydrogen production from KBH_4_ hydrolysis. The effects of the molar ratio of ISOBAM-104 to metal ion, concentration of metal ion, and molar ratio of Co/Fe were investigated. The as-synthesized ISOBAM-104 stabilized Co_10_Fe_90_ colloidal catalysts possess an unexpected catalytic activity for hydrogen production from KBH_4_ hydrolysis at room temperature. The activation energy of the as-prepared Co_10_Fe_90_ colloidal catalysts towards KBH_4_ hydrolysis was calculated by the Arrhenius formula. In addition, the electronic property of metal atoms was investigated based on the DFT calculations.

## 2. Experimental Section

### 2.1. Materials

Potassium borohydride (KBH_4_), sodium hydroxide (NaOH), iron nitrate nonahydrate (Fe(NO_3_)_3_·9H_2_O) and cobalt nitrate hexahydrate (Co(NO_3_)_2_·6H_2_O) were purchased from Sinopharm Chemical Reagent Co., Ltd., Shanghai, China. ISOBAM-104 (NO. 52032-17-4, Appendix A) was purchased from Kuraray Co., Ltd., Tokya, Japan. The deionized water was produced via a PINGGUAN ultrapure water purification system (Wuhan, China).

### 2.2. Preparation of Co/Fe Colloidal Catalysts and Hydrogen Generation

Firstly, certain concentrations of Co(NO_3_)_2_·6H_2_O and Fe(NO_3_)_3_·9H_2_O solution were mixed together in a three-neck flask (Appendix A). Next, a certain amount of ISOBAM-104 was added into the flask and it was then filled with deionized water to 50 mL. After that, the mixed solution was continuously stirred for 24 h at room temperature. Subsequently, the configured KBH_4_ and NaOH solution were rapidly added into the above solution to obtain ISOBAM-104 protected Co/Fe BNPs.

The influence of the molar ratio of ISOBAM-104 to metal ion concentration (denoted as *R_ISO_*, from 10 to 80), metal ion concentration (from 0.6 to 1.5 mM), and chemical composition (Fe, Co_10_Fe_90_, Co_30_Fe_70_, Co_50_Fe_50_, Co_70_Fe_30_, Co_90_Fe_10_, and Co) were investigated. The detailed batch compositions are shown in the Appendix A. The volume of generated H_2_ was measured by an electronic balance, which was automatically recorded based on the displacement level of water every two seconds. During this process, the generated gas was passed through a trap containing concentrated H_2_SO_4_ to remove H_2_O and any NH_3_ that might have been generated. The rate of hydrogen generation (*k*, mL-H_2_·min^−1^) could be obtained from the slope of H_2_ volume–time curve in the initial stage of the reaction.

The catalytic activity (mL-H_2_·min^−1^·g-cat^−1^) could be calculated by the ratio of the hydrogen generation rate (*k*) to the mass of catalyst (*m*). It should be noted that the ISOBAM-104 used in this work contains the NH_4_^+^ group, which also possesses a catalytic effect for KBH_4_ hydrolysis [19,37]. Therefore, under the same condition, the catalytic activities of ISOBAM-104 stabilized Co/Fe colloidal catalysts and ISOBAM-104 (NH_4_^+^ group) were measured. The intrinsic catalytic activity value of Co/Fe colloidal catalysts were obtained by subtracting the value of ISOBAM-104 from that of ISOBAM-104 stabilized catalysts. All the catalytic experiments were repeated no less than three times under the identical condition. The average values, which were normalized to mL-H_2_ min^−1^ g-Co^−1^, were used to determine the catalytic activity (detailed calculation procedures are provided in the Appendix A).

### 2.3. Material Characterization

UV-vis absorption spectra were recorded at 200–800 nm by a Shimadzu UV-2550 spectrophotometer (Shimadzu Company, Kobe, Japan). Transmission electron microscopy (TEM) and high-resolution transmission electron microscopy (HRTEM) images were collected by using a JEM-2100F (JEOL Company, Tokyo, Japan). The average size of the nanoparticles in each sample was estimated by measuring at least 200 particles from different parts of the grid. Fourier transform infrared (FTIR) spectra were obtained on a FTIR spectrometer (VERTEX 70, Bruker Corporation, Karlsruhe, Germany), and the samples were embedded in KBr pellet. X-ray photoelectron spectroscopy (XPS) measurements were performed on a VG MultiLab 2000 instrument (Thermo Electron Corporation, Massachusetts, USA) equipped with a 300 W Al Kα excitation source. The obtained XPS spectra were calibrated using a reference energy of 284.6 eV for the C 1s level and analyzed by Avantage software.

### 2.4. Density Functional Theory (DFT) Calculation

The spin-polarized density functional theory (DFT) calculations were carried out using a generalized gradient approximation (GGA) with Perdew–Burke–Ernzerhof (PBE) exchange-correlation functional [38], as implemented in the DMol^3^ package (BIOVIA Company, San Diego, CA, USA) [39]. The double numerical basis set and polarization functions (DNP) were carried out to describe the valence electrons, and an electron relativistic core treatment was used to perform full optimization of the investigated cluster model of Co_6_Fe_49_ BNP without symmetry constraint. The convergence criteria were set to medium quality with a tolerance for the self-consistent field (SCF), optimization energy, maximum force, and maximum displacement of 10^−5^ Ha, 2 × 10^−5^ Ha, 0.004 Ha/Å and 0.005 Å, respectively. The charge analysis was performed on the basis of the Mulliken population distribution scheme [40,41].

## 3. Results and Discussion

### 3.1. Effect of R_ISO_ on the Activity of Co/Fe Colloidal Catalysts

To explore the optimized reaction condition, the effect of *R_ISO_* on the preparation and catalytic activity of the Co/Fe BNPs was systematically investigated. The TEM images (Figure 1) and size distribution histograms (Appendix A) indicate that the average particle sizes of Co_50_Fe_50_ BNPs are about 4.6 nm (*R_ISO_* = 10), 3.7 nm (*R_ISO_* = 30), 3.2 nm (*R_ISO_* = 50), and 2.3 nm (*R_ISO_* = 80), respectively. Obviously, the average particle size decreases with the increase of *R_ISO_* value, which may be ascribed to the fact that the increase of the protective agents could provide a large number of −COO^−^ and −NH_2_ groups to prevent the agglomeration of particles. Figure 2 displays the catalytic activities of the obtained Co_50_Fe_50_ colloidal catalysts for hydrogen production at different *R_ISO_*. It can be clearly observed that the Co_50_Fe_50_ colloidal catalysts with *R_ISO_* = 50 possess a higher catalytic value (17,500 mL-H_2_ min^−1^ g-Co^−1^) than those synthesized at *R_ISO_* = 10, 30, and 80 (6800, 6600, and 5500 mL-H_2_ min^−1^ g-Co^−1^, respectively). This result may be attributed to the fact that Co_50_Fe_50_ nanoparticles cannot receive effective protection at low *R_ISO_* and are prone to agglomeration, leading to a low catalytic activity. Comparatively, when *R_ISO_* was superfluous, the surface of the nanoparticles would be covered by ISOBAM-104, resulting in the decrease of active sites and catalytic activity [28]. Thus, based on the above results, the Co_50_Fe_50_ catalysts with moderate particle size and high catalytic activity could be synthesized when *R_ISO_* = 50.

### 3.2. Effect of Metal Ion Concentration on the Activity of Co/Fe Colloidal Catalysts

The effect of ion concentration on the preparation and catalytic activity of Co_50_Fe_50_ colloidal catalysts was also investigated. TEM morphologies and size distribution histograms of the as-prepared Co_50_Fe_50_ BNPs are presented in Figure 3 and Appendix A. The average particle sizes are about 2.3, 3.2, 2.6, and 3.4 nm at the metal ion concentrations of 0.6, 0.9, 1.2, and 1.5 mM, respectively. It is found that the metal ion concentration exerts a significant influence on the particle size of the obtained catalysts. Although the Co_50_Fe_50_ BNPs with the smaller particle sizes are obtained at the metal ion concentrations of 0.6 mM, the low concentration of metal ion impedes the large-scale preparation of catalysts. Hence, the concentration of metal ion is set as 1.2 mM in the following discussion.

### 3.3. Effect of Chemical Composition on the Activity of Co/Fe Colloidal Catalysts

The UV-vis spectra of the obtained Co/Fe BNPs with various compositions are shown in Appendix A. It was found that no surface plasma resonance peak of Fe or Co nanoparticles could be detected, which agrees with the previous reports [26,27,42]. The spectra of the dispersed Co/Fe nanoparticles BNPs with a featureless absorbance were located between the spectra of single Co and Fe nanoparticles, exhibiting a featureless absorbance. These obvious differences of the absorbance at various Fe content suggest the formation of alloy-structured Co/Fe BNPs. Figure 4 presents the TEM images of the obtained Co/Fe BNPs at various Co/Fe atomic ratios. It can be clearly seen that the particles possessed a sphere-like morphology. The average sizes of ISOBAM-104 stabilized Fe, Co_10_Fe_90_, Co_30_Fe_70_, Co_50_Fe_50_, Co_70_Fe_30_, Co_90_Fe_10_, and Co colloidal catalysts are respectively about 3.0, 3.2, 2.6, 2.6, 2.2, 2.5, and 1.8 nm (Appendix A). The corresponding catalytic activities of the above colloidal catalysts are displayed in Figure 5. By comparison, the above-mentioned Co/Fe BNPs presented a superior catalytic activity than that of Co or Fe MNPs. More importantly, the catalytic activity of the Co_10_Fe_90_ colloidal catalysts reaches up to 37,900 mL-H_2_ min^−1^ g-Co^−1^, which is about 5 and 4 times higher than that of Fe (7400 mL-H_2_ min^−1^ g-Fe^−1^) and Co (9600 mL-H_2_ min^−1^ g-Co^−1^), respectively. Base on the above results, the desirable Co/Fe colloidal catalysts with high catalytic performance can be synthesized at the chemical composition of Co_10_Fe_90_, *R_ISO_* = 50, and ion concentrations of 1.2 mM.

The structure of the obtained Co_10_Fe_90_ colloidal catalysts was further characterized by the high-resolution transmission electron microscope (HRTEM). As shown in Figure 6, the interplanar spacings of the four individual randomly-chosen Co/Fe BNPs are measured as 0.168, 0.172, 0.174, and 0.169 nm, respectively. These values are inconsistent with the theoretical interplanar spacing values of Co and Fe (Appendix A). However, it is worth noting that this measured interplanar spacing located between the interplanar distance of Co (200) and Fe (200) (Appendix A), suggests the alloy structure of the formed Co/Fe BNPs.

In order to understand the protecting role of ISOBAM-104 in the catalysts stabilization, the FTIR spectra of ISOBAM-104 stabilized Co/Fe catalysts, ISOBAM-104, Co(NO_3_)_2_, and Fe(NO_3_)_3_ are displayed in Appendix A. The absorption peak at 1400, 1680, 2300, and 3400 cm^−1^, respectively, correspond to the stretching vibration of –OH, –COOH, –CO_2_, and the –NH_2_ group of ISOBAM-104. By comparison, it can be clearly seen that the –COOH group of ISOBAM-104 disappeared, while the –OH and –NH_2_ group still appeared in the ISOBAM-104 stabilized Co/Fe catalysts, demonstrating that the –NH_2_ group in ISOBAM-104 should play a protective role on the as-prepared metal catalyst [18].

### 3.4. Kinetic Study and Catalytic Mechanism of Co/Fe Colloidal Catalysts

To calculate the apparent activation energy (*E_a_*), the catalytic performance of Co_10_Fe_90_ colloidal catalysts were evaluated under the perturbation of the reaction temperature. As shown in Appendix A, it can be seen that the catalytic activity of the Co_10_Fe_90_ colloidal catalysts increases from 8400 to 15,200 mL-H_2_ min^−1^ g-catalyst^−1^ as the temperature increases from 293 to 308 K. The *E_a_* is calculated by using the Arrhenius method [43]. As shown in Figure 7, the slope of the linear curve between the natural logarithm of catalytic activity and the reciprocal of temperature is −*E_a_*/R, where R is the universal gas constant. The calculated *E_a_* of Co_10_Fe_90_ colloidal catalysts is 14.6 ± 0.7 kJ mol^−1^, which is much lower than most of the reported metal-based catalysts (Table 1). Interestingly, the corresponding catalytic activity of the Co_10_Fe_90_ colloidal catalysts is much higher than these metal-based catalysts. Thus, it can be confirmed that the excellent catalytic activity of Co_10_Fe_90_ colloidal catalysts is closely related to the lower activation energy towards KBH_4_ hydrolysis.

An XPS measurement was subsequently carried out to clarify the elemental composition and valence state of the Co_10_Fe_90_ BNPs. In Appendix A, the element of Co, Fe, O, N, C, and B are detected in the obtained Co/Fe colloidal catalysts. The high-resolution XPS spectra of Co 2p (Appendix A) shows that the electron binding energy of Co^0^ 2p_3/2_ (776.0 eV) is about 2.3 eV lower than that of the bulk Co (778.3 eV), indicating a negatively-charged characteristic of Co atoms in Co_10_Fe_90_ BNPs. Meanwhile, the electron binding energy of Fe^0^ 2p_3/2_ (708.5 eV) was about 1.8 eV higher than that of the bulk Fe (706.7 eV), suggesting that the Fe atoms were positively charged (Appendix A). The negative shift of the Co^0^ 2p_3/2_ binding energy and positive shift of the Fe^0^ 2p_3/2_ binding energy might be ascribed to the electron charge transfer occurring between Fe and Co atoms [23,24,26,50,51]. To further confirm the electron transfer effect, DFT calculations were employed to investigate the electronic states of each atom in the Co_6_Fe_49_ alloy nanoparticles [52]. As shown in Figure 8a, the Co atoms are negatively charged (−0.091 eV), while the Fe atoms are positively charged (0.029 or 0.021 eV), which is matched well with the above XPS result. Based on above discussions and the related literature [23,27], a plausible mechanism for the high catalytic performance of Co/Fe colloidal catalysts could be proposed. Due to the charge transfer between Fe atoms and Co atoms (Figure 8b), the negatively charged Co atoms are conducive to the fracture of H−O bonds in H_2_O molecules, and the positively charged Fe atoms could promote the B−H bond breaking in KBH_4_ molecules. As a result, the catalytic activity of Co/Fe colloidal catalysts for KBH_4_ hydrolysis could be markedly enhanced under the synergistic effect of Fe and Co atoms.

## 4. Conclusions

In summary, the ISOBAM-104 stabilized Co/Fe colloidal catalysts are successfully synthesized for hydrogen generation by a simple co-reduction method via using ISOBAM-104 as a protective agent, and Co(NO_3_)_2_·6H_2_O, Fe(NO_3_)_3_·9H_2_O, and KBH_4_ as starting materials. The catalytic activities of the obtained Co/Fe colloidal catalysts could reach up to 37,900 mL-H_2_ min^−1^ g-Co^−1^ at the chemical composition of Co_10_Fe_90_, *R_ISO_* = 50, and ion concentrations of 1.2 mM, which is superior to their corresponding monometallic nanoparticles. The excellent catalytic activity of Co_10_Fe_90_ colloidal catalysts is mainly attributed to their lower activation energy towards KBH_4_ hydrolysis, and the charge transfer effect between Fe and Co atoms. This finding could provide a deeper insight for developing the economic, highly active, and recyclable bimetallic catalysts.

## Figures and Tables

**Figure 1 nanomaterials-12-02998-f001:**
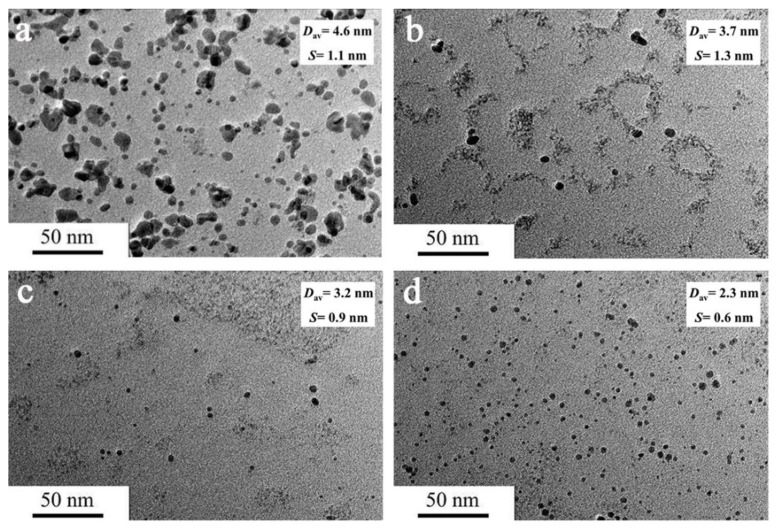
TEM images of Co_50_Fe_50_ colloidal catalysts with various *R_ISO_* ([Co^2+^ + Fe^3+^] = 0.9 mM; *R_ISO_* = 10 (**a**), 30 (**b**), 50 (**c**), and 80 (**d**)). (D_av_: average particle size; S: standard deviation).

**Figure 2 nanomaterials-12-02998-f002:**
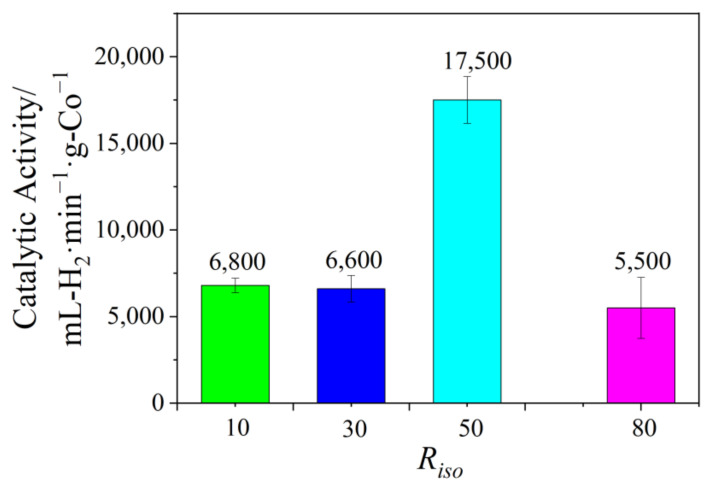
Comparison of catalytic activity of Co_50_Fe_50_ colloidal catalysts with varied *R_ISO_* ([Co^2+^ + Fe^3+^] = 0.9 mM).

**Figure 3 nanomaterials-12-02998-f003:**
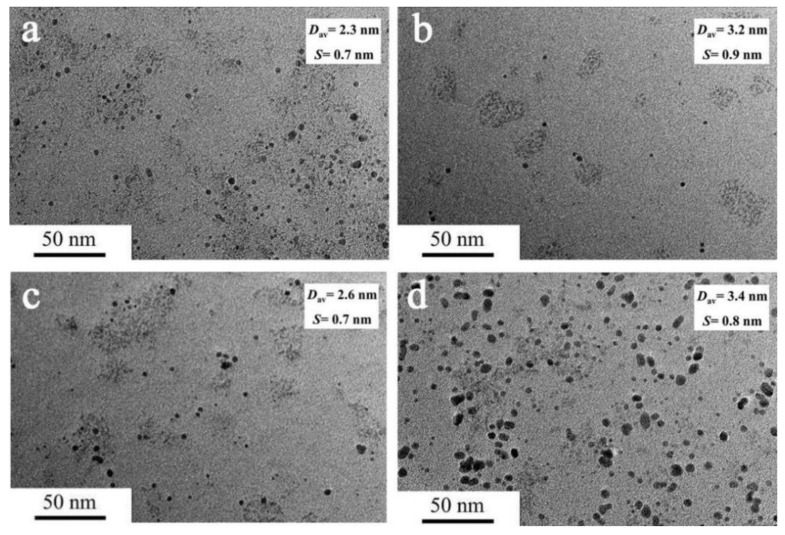
TEM images and size distribution histograms of Co_50_Fe_50_ colloidal catalysts synthesized with different ion concentrations ([Co^2+^ + Fe^3+^] = 0.6 (**a**), 0.9 (**b**), 1.2 (**c**), and 1.5 (**d**) mM). (D_av_: average particle size; S: standard deviation).

**Figure 4 nanomaterials-12-02998-f004:**
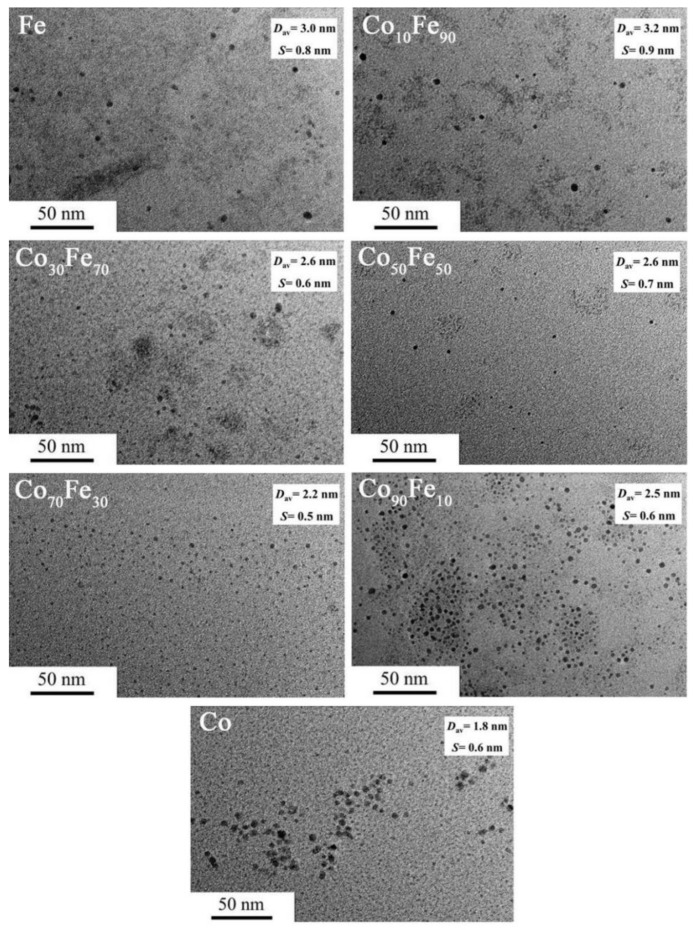
TEM images of Co/Fe colloidal catalysts synthesized with various chemical compositions (*R_ISO_* = 50, [Co^2+^ + Fe^3+^] = 1.2 mM). (D_av_: average particle size; S: standard deviation).

**Figure 5 nanomaterials-12-02998-f005:**
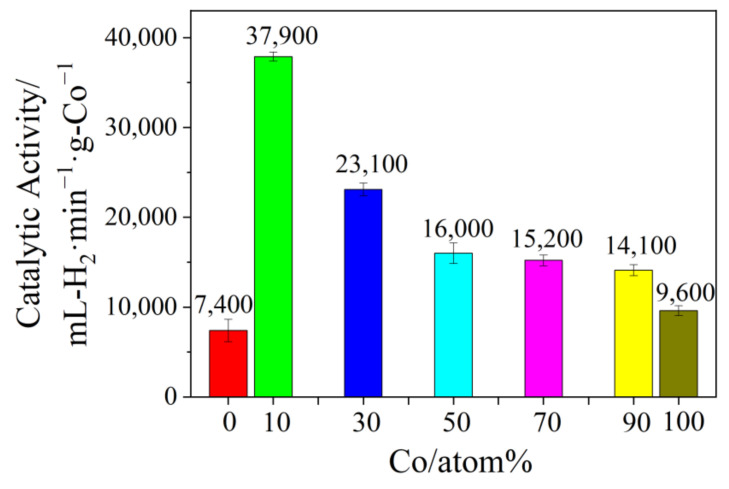
Comparison of catalytic activity of Co/Fe colloidal catalysts with various chemical compositions (*R_ISO_* = 50, [Co^2+^ + Fe^3+^] = 1.2 mM).

**Figure 6 nanomaterials-12-02998-f006:**
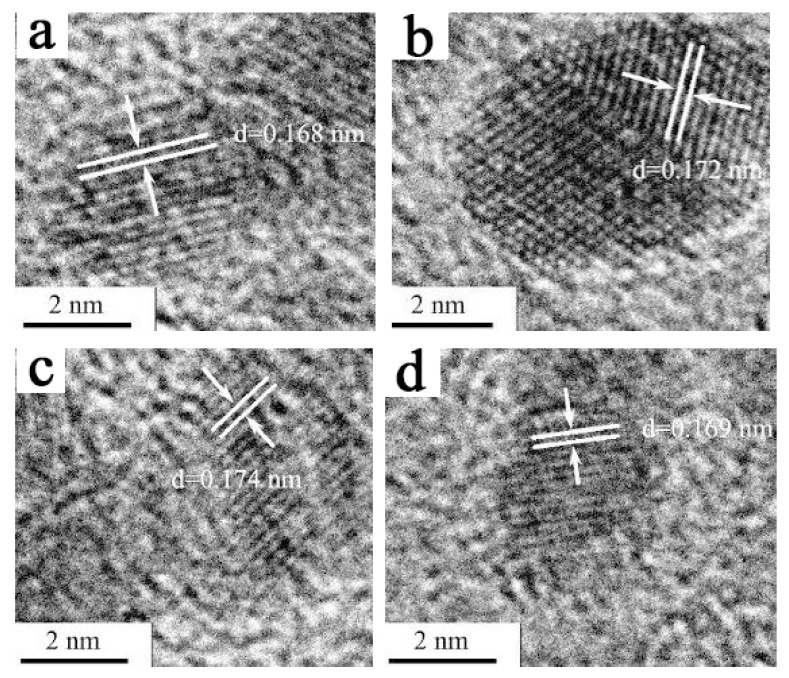
HRTEM images (**a**–**d**) of Co_10_Fe_90_ colloidal catalysts (*R_ISO_* = 50, [Co^2+^ + Fe^3+^] = 1.2 mM). (HRTEM images of a–d correspond to four individual randomly-chosen Co/Fe bimetallic nanoparticles.).

**Figure 7 nanomaterials-12-02998-f007:**
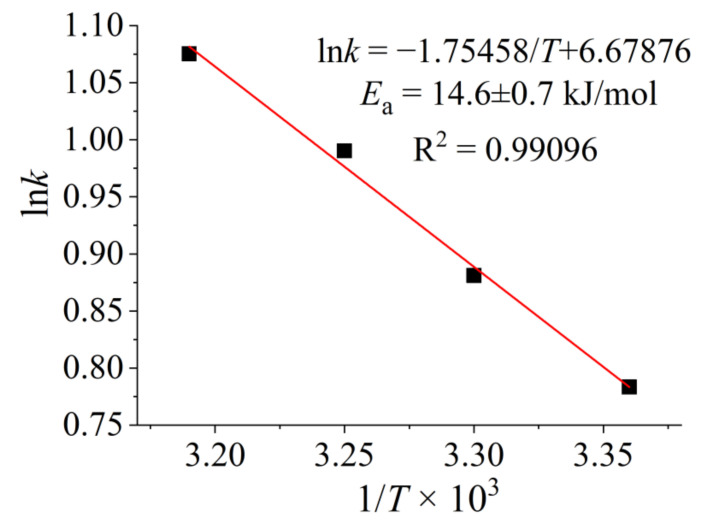
The apparent activation energy (*E_a_*) of Co_10_Fe_90_ colloidal catalysts for KBH_4_ hydrolysis at 293−308 K.

**Figure 8 nanomaterials-12-02998-f008:**
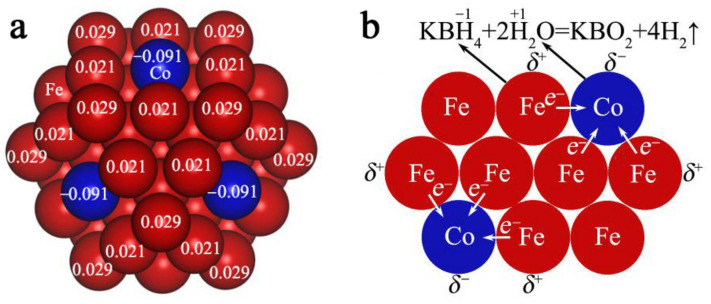
Catalytic mechanism. (**a**) DFT calculations of the electronic structure of Co_6_Fe_49_ nanoparticles (red, Fe; and blue, Co). (**b**) Schematic illustration of the possible electron charge transfer effects between Co and Fe atoms in the Co_6_Fe_49_ nanoparticles.

**Table 1 nanomaterials-12-02998-t001:** Comparison of the apparent activation energy between the Co_10_Fe_90_ colloidal catalysts and other catalysts in the previously reported literature.

Catalyst	Reactant	Activation Energy (kJ mol^−1^)	Catalytic Activity (mL-H_2_ min^−1^ g-cat.^−1^)	Reference
Co/Fe	KBH_4_	14.6	37,900	Present work
Ni	KBH_4_	41.3	12,400	[18]
Rh/Ni	KBH_4_	47.2	11,580	[27]
Co-O-P	NaBH_4_	63	4850	[44]
Ag/Ni	NaBH_4_	16.2	2333	[45]
Co-Ni-P	NaBH_4_	31.2	6681	[46]
Co-B	NaBH_4_	37.57	2649	[47]
Co-B	NaBH_4_	30	5310	[48]
CoO−Co_2_P	NaBH_4_	27.4	3940	[49]

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
