# Peer review of "Excellent Catalytic Performance of ISOBAM Stabilized Co/Fe Colloidal Catalysts toward KBH4 Hydrolysis"

_nanomaterials, 2022, doi:10.3390/nano12172998_

Round 1

Reviewer 1 Report

The work presented in this ms could be interesting to researchers working in the area of catalysis. However, I have several comments that came across while reviewing this ms. They are given below:

1-     The calculation details section is poorly presented. Appropriate references are not given. For example, the literature for the PBE functional is not cited. What specific PAW potential was used for specific atoms is not mentioned, even though there are many variants of the PAW potential known. The cutoff for optimization energy was in Ha, or eV? Author ares required to check all details from manual of the code before writing them in a paper!!

2-     What is a VAS code? I never know it is existing at all!!! Should it be something else?

3-     The FTIR spectral peaks are given on page 7. However, the discussion is poorly presented. I am not sure how the catalyst modified the frequencies? It is not clear whether there was a red- or blue-shift in the stretching frequencies of some modes that are affected by the catalyst?

4-     How was the activation energy of the systems in Table 1 obtained? Were they reproduced by DFT calculations? I ask the authors to provide the reaction pathways for some systems, describing the activation energy. This could be done with DFT calculations. Otherwise, I am not sure how reliable are the data presented in Table 1.

5-     The paper is not well written. Significant rewriting is necessary. Otherwise, authors should consult their English-speaking colleagues and revise the paper.

6-     Background references are not adequately given – many interesting papers related to this work, which are reported by others, are missing. A mere google search can assist the authors to get many of them. 

Reviewer 2 Report

The manuscript titled “Excellent catalytic performance of ISOBAM stabilized Co/Fe colloidal catalysts toward KBH4 hydrolysis” presented by Keke Guan , Qing Zhu, Zhong Huang, Zhenxia Huang, Haijun Zhang, Junkai Wang, Quanli Jia and Shaowei Zhang deals with the catalytic measurements and characterization of COFe-ISOBAM-104. However, there are major issues that could be clarified:

  • The abstract should not content the abbreviations, for example, XPS. As well MNPs and DFT are not used in abstract after first mention.
  • The introduction contents the statement about the size effect in catalysis. (Previous researches reported that the catalytic activity of metal particles is closely related to the particle size. [17-18] With decreasing the particle size of metal catalysts, the catalytic activity could be significantly enhanced. Over the past decade, many researches have been committed to explore various metal catalysts with small size. Nevertheless, the issue for the protection and the large-scale production of the metal particles still need to be addressed.[20]). The authors discuss only the KBH4 hydrolysis, but the readers could interpreter the statement made for all catalysis reaction. Please, re-write this part.
  • The sentence “The influence of the molar ratio of ISOBAM-104 to metal ion concentration (denoted as RISO, from 10 to 80), metal ion concentration (from 0.6 to 1.5 mM), and chemical composition (Fe, Co10Fe90, Co30Fe70, Co50Fe50, Co70Fe30, Co90Fe10, and Co) were investigated, as shown in the Table S1.” Should be re-written for more understandable way. Table S1 does not content the information about investigation.
  • The description of XPS setup demands some corrections. The charge correction, the software used. Could you clarify the phrase “with 300 W Al Kα as the excitation source”?
  • The information about catalytic measurements is absent. The catalytic weight, the hydrogen detection, and measurements of hydrogen yield. Etc.
  • Could author explain how the variation of RISO (the molar ratio of ISOBAM-104 to metal ion concentration) differs from the variation of total metal ion concentration for Co50Fe50? Could you calculate RISO for metal ion concentration for Co50Fe50 (0.6, 0.9, 1.2, and 1.5 mM)? Please, add the molar concentration of ISOBAM-104 for all catalysts in Table S1.
  • The analysis of XPS Co2p core-level spectrum is incorrect. The incorrect background subtraction leads to introduction of fantom peaks at 776 eV that authors interpreter as negative charging of Co atoms. It is absolutely wrong interpretation. One can see if the BG subtraction is correctly done the correct binding energy of Co2p3/2 is about 780 eV corresponding to Co2+. No charge transfer is observed. The authors should provide the correct fitting of the new-measured Co2p and Fe2p core-level spectra with high signal-to-noise ratios.
  • Provide the [Co]/[Fe] ratios measured by XPS for all catalysts.
  • The other conclusions based charge transfer are doubt.
  • Based on figure S2 one can conclude that the hydrogen yield evaluation approach is non-classic. The use of gas chromatography is more suitable and contemporary.
  • The main question concerns to the rate of H2 yield. The authors should add the information about the rates of H2 yield for the similar process and different catalysts. It is enough to extend Table 1.
  • Please, use the correct terms: UV-vis (not vis), FTIR, eV, etc.

Reviewer 3 Report

This paper presents interesting and useful results related to design and characterization of ISOBAM stabilized Co/Fe colloidal catalysts of KBH4 hydrolysis. However, it should be revised to bring it to standards of international publication.

1.      Methodics of samples preparation for XPS studies are to be given. Since in Fig. S9 b apparently bands corresponding to Co 2+ and 3+ states  dominate, it is clear that samples contacted with air during supporting on samples holder without any further  pretreatment. In this case reliable estimation of the surface composition of alloys nanoparticles is doubtful.

2.      In Fig. S9 spectra of Fe2p  are absent

3.      Equations used for  calculation of catalytic activity and reaction rate constant are to be given.

4.      English is to be polished , both grammar, syntax and terms usage. Typical examples are present in Introduction , such as “systematacially”, “the electronic property between Fe and Co atoms”, etc.  

Round 2

Reviewer 1 Report

The authors of this work have revised their paper, and the changes are highlighted in the revised ms. I still have some concerns.

1 – Authors wrote this in the computational section: “Double numerical basis set and polarization functions were carried out to describe the valence electrons, and an electron relativistic core treatment was used to perform full optimization of the investigated cluster model of Co6Fe49 BNP without symmetry constraint. “ I had previously used Dmol3, and I know that each basis set the code provides do have a name. Why then the name is not used?

2 - The convergence criteria were set to medium quality with a tolerance for self-consistent field (SCF), optimization energy, maximum force, and maximum displacement of 10−5 Ha, 2 × 10−5 Ha, 0.004 Ha/Å and 0.005 Å, respectively.

In most of the papers reported, the cutoff for energy is generally given from 10-5 to 10-8, in unit of eV. However, the authors have mentioned the unit in Hartree, Ha. I doubt if it is correct??

3 – Although the catalytic performance of a material is discussed in this work, I cannot see the reaction pathway. I mentioned this in my previous review, but authors have failed to provide any detail on this. While simulations were done with Dmol3, why then the reaction pathway to discover the transition state is not explored? This should have given in the form of a plot.

Reviewer 2 Report

The manuscript could be accepted for publication.

Round 3

Reviewer 1 Report

Authors have revised their paper, including the computational section. This latter section reads: 

Density Functional Theory (DFT) Calculation DFT calculations were carried out using spin-polarization DFT/GGA with the Perdew-Burke-Ernzerhof (PBE) exchange correlation functional,[38] as implemented in the DMol3 package (BIOVIA company, San Diego, CA, USA).[39] Double numerical basis set and polarization functions (DNP) were carried out to describe the valence electrons, and an electron relativistic core treatment was used to perform full optimization of the investigated cluster model of Co6Fe49 BNP without symmetry constraint. The convergence criteria were set to medium quality with a tolerance for self-consistent field (SCF), optimization energy, maximum force, and maximum displacement of 10−5 Ha, 2 × 10 −5 Ha, 0.004 Ha/Å and 0.005 Å, respectively. Charge analysis was performed on the basis of the Mulliken population distribution scheme .[40,41] 

1- I have never seen terms like "... using spin-polarization DFT/GGA " in literature. What is often appeared in research papers is the term "spin-polarized DFT calculations". The next question is then: Why was spin-polarization taken into account? What were the magnetic moments  of spin-polarized constituent? These essential features are not discussed, and should be included.  

2- Many terms such as "DFT", "GGA", among others, are not defined in the paper, but used. This can greatly confuse readers. 

3- There are many grammatical errors scattered throughout the ms. The should be read by a native English writer
